# Bayesian Persuasion for Algorithmic Recourse

**Keegan Harris**
Carnegie Mellon University
keeganh@cmu.edu

**Valerie Chen**
Carnegie Mellon University
valeriechen@cmu.edu

**Joon Sik Kim**
Carnegie Mellon University
joonkim@cmu.edu

**Ameet Talwalkar**
Carnegie Mellon University
talwalkar@cmu.edu

**Hoda Heidari**
Carnegie Mellon University
hheidari@cmu.edu

**Zhiwei Steven Wu**
Carnegie Mellon University
zstevenwu@cmu.edu

## Abstract

When subjected to automated decision-making, decision subjects may strategically modify their observable features in ways they believe will maximize their chances of receiving a favorable decision. In many practical situations, the underlying assessment rule is deliberately kept secret to avoid gaming and maintain competitive advantage. The resulting opacity forces the decision subjects to rely on *incomplete information* when making strategic feature modifications. We capture such settings as a game of *Bayesian persuasion*, in which the decision maker offers a form of recourse to the decision subject by providing them with an action recommendation (or *signal*) to incentivize them to modify their features in desirable ways. We show that when using persuasion, the decision maker and decision subject are *never worse off* in expectation, while the decision maker can be *significantly better off*. While the decision maker's problem of finding the optimal Bayesian incentive-compatible (BIC) *signaling policy* takes the form of optimization over infinitely-many variables, we show that this optimization can be cast as a linear program over finitely-many regions of the space of possible assessment rules. While this reformulation simplifies the problem dramatically, solving the linear program requires reasoning about exponentially-many variables, even in relatively simple cases. Motivated by this observation, we provide a polynomial-time approximation scheme that recovers a near-optimal signaling policy. Finally, our numerical simulations on semi-synthetic data empirically demonstrate the benefits of using persuasion in the algorithmic recourse setting.

## 1 Introduction

High-stakes decision-making systems increasingly utilize data-driven algorithms to assess individuals in such domains as education [31], employment [5, 36], and lending [24]. Individuals subjected to these assessments (henceforth, decision subjects) may strategically modify their observable features in ways they believe maximize their chances of receiving favorable decisions [21, 9]. The decision subject often has a set of actions/interventions available to them. Each of these actions leads to some measurable effect on their observable features, and subsequently, the decision they receive. From the decision maker's perspective, some of these actions may be more desirable than others. Consider credit scoring as an example.[1] Credit scores predict how likely an individual applicant

---

[1]Other examples of strategic settings which arise as a result of decision-making include college admissions, in which a college/university decides whether or not to admit a prospective student, hiring, in which a company decides whether or not to hire a job applicant, and lending, in which a banking institution decides to accept or reject someone applying for a loan. Oftentimes, the decision maker is aided by automated decision-making tools in these situations (e.g., [31, 38, 24]).

36th Conference on Neural Information Processing Systems (NeurIPS 2022).

is to pay back a loan on time. Financial institutions regularly utilize credit scores to decide whether to offer applicants their financial products and determine the terms and conditions of their offers (e.g., by setting the interest rate or credit limit). Applicants regularly attempt to improve their scores given their (partial) knowledge of credit scoring instruments. For instance, a business applying for a loan may improve its score by paying off existing debt or cleverly manipulating its financial records to appear more profitable. While both of these interventions may improve credit score, the former is more desirable than the latter from the perspective of the financial institution offering the loan. The question we are interested in answering in this work is: *how can the decision maker incentivize decision subjects to take such beneficial actions while discouraging manipulations?*

The strategic interactions between decision-making algorithms and decision subjects has motivated a growing literature known as *strategic learning* (see e.g., [18, 11, 40, 29, 19]). While much of the prior work in strategic learning operates under the assumption of full transparency (i.e., the assessment rule is public knowledge), we consider settings where the full disclosure of the assessment rule is not viable. In many real-world situations, revealing the exact logic of the decision rule is either infeasible or irresponsible. For instance, credit scoring formulae are closely guarded trade secrets, in part to prevent the risk of default rates surging if applicants learn how to manipulate them. Moreover, the underlying decision rule is often *fixed* ahead of time due to institutional structuring. In our credit scoring example, one department of the bank may be in charge of determining the threshold on the credit assessment, while another department may be in charge of offering recourse.[2]

In such settings, the decision maker may still have a vested interest in providing *some* information about the decision rule to decision subjects in order to provide a certain level of *transparency* and *recourse*. In particular, the decision maker may be legally obliged, or economically motivated, to guide decision subjects to take actions that improve their underlying qualifications. To this end, instead of fully revealing the assessment rule, the decision maker can *recommend actions* for decision subjects to take. Of course, such recommendations need to be chosen carefully and credibly; otherwise, self-interested decision subjects may not follow them or may utilize the recommendations to find pathways for manipulation.

We study a model of strategic learning in which the underlying assessment rule is not revealed to decision subjects. Our model captures several key aspects of the setting described above: First, even though the assessment rule is not revealed to the decision subjects, they often have *prior knowledge* about what the rule may be. Secondly, when the decision maker provides recommendations to decision subjects on which action to take, the recommendations should be *compatible with the subjects' incentives* to ensure they will follow the recommendation. Finally, our model assumes the decision maker discloses how they generate recommendations for recourse—an increasingly relevant requirement under recent regulations (e.g., [10]).

Utilizing our model, we aim to design a mechanism for a decision maker to provide recourse to a decision subject who has incomplete information about the underlying assessment rule. We assume the assessment rule makes *predictions* about some future outcome of the decision subject (e.g., whether they will pay back a loan in time if granted one). Before the assessment rule is trained (i.e., before the model parameters are fit), the decision maker and decision subject have some *prior belief* about the realization of the assessment rule. This prior represents the "common knowledge" about the importance of various observable features for making accurate predictions. After training, the assessment rule is revealed to the decision maker, who then recommends an action for the decision subject to take, based on their pre-determined *signaling policy*. Upon receiving this action recommendation, the decision subject updates their belief about the underlying assessment rule. They then take the action which they believe will maximize their utility (i.e., the benefit from the decision they receive, minus the cost of taking their selected action) in expectation. Finally, the decision maker uses the assessment rule to make a prediction about the decision subject.

The interaction described above is an instance of *Bayesian persuasion*, a game-theoretic model of information revelation originally due to Kamenica and Gentzkow. The specific instance of Bayesian persuasion we consider in this work is summarized in Figure 1.

---

[2]Similar logic applies to other examples of strategic settings including: college admissions, in which someone associated with the university may have the ability to offer advice to applicants, but does not have the ability to unilaterally change the underlying assessment rule, or hiring, where a recruiter for a company may have knowledge of the factors the company uses to make hiring decisions, but may not be able to change this criteria or reveal it to job applicants.

| Interaction protocol between the decision maker and decision subject |
| :--- |

1. The decision maker and decision subject initially have some prior/belief about the assessment rule that will be trained.
2. Before training, the decision maker commits to a signaling policy. After training, the assessment rule is revealed to the decision maker.
3. The decision maker then uses their signaling policy and knowledge of the assessment rule to recommend an action for the decision subject to take.
4. The decision subject updates their belief given the recommendation, and chooses an action that they believe maximizes their utility.
5. The decision subject receives a prediction through the assessment rule.

Figure 1: Summary of the setting we consider.

**Our contributions.** Our central conceptual contribution is to cast the problem of offering recourse under partial transparency as a game of Bayesian persuasion. Our key technical contributions consist of comparing optimal action recommendation policies in this new setup with two natural alternatives: (1) fully revealing the assessment rule to the decision subjects, or (2) revealing no information at all about the assessment rule. We provide new insights about the potentially significant advantages of action recommendation over these baselines, and offer efficient formulations to derive the optimal recommendations. More specifically, our analysis offers the following takeaways:

1. Using tools from Bayesian persuasion, we show that it is possible for the decision maker to provide incentive-compatible action recommendations that encourage rational decision subjects to modify their features through beneficial interventions. While the decision maker and decision subjects are never worse off in expectation from using optimal incentive-compatible recommendations, we show that situations exist in which the decision maker is *significantly better off* in expectation utilizing the optimal signaling policy (as opposed to the two baselines, Section 3).
2. We derive the optimal signaling policy for the decision maker. While the decision maker's optimal signaling policy initially appears challenging to compute (as it involves optimizing over *continuously-many* variables), we show that the problem can naturally be cast as a linear program defined in terms of a finite set of variables. However, solving this linear program may require reasoning about exponentially-many variables. Motivated by this observation, we provide a polynomial-time algorithm to approximate the optimal signaling policy (Section 4).
3. We empirically evaluate our persuasion mechanism on semi-synthetic data based on the Home Equity Line of Credit (HELOC) dataset, and find that the optimal signaling policy performs significantly better than the two natural alternatives across a wide range of instances (Section 5).

## 1.1 Related work

**Strategic responses to unknown predictive models.** To the best of our knowledge, our work is the first to use tools from persuasion to model the strategic interaction between a decision maker and strategic decision subjects when the underlying predictive model is not public knowledge. Several prior works have addressed similar problems through different models and techniques. For example, Akyol et al. [1] quantify the "price of transparency", a quantity which compares the decision maker's utility when the predictive model is fully known with their utility when the model is not revealed to the decision subjects. Tsirtsis and Rodriguez [41] study the effects of counterfactual explanations on strategic behavior. Ghalme et al. [17] compare the prediction error of a classifier when it is public knowledge with the error when decision subjects must learn a version of it, and label this difference the "price of opacity". They show that small errors in decision subjects' estimates of the true underlying model may lead to large errors in the performance of the model. The authors argue that their work provides formal incentives for decision makers to adopt full transparency as a policy. Our work, in contrast, is based on the observation that even if decision makers are willing to reveal their models, legal requirements, privacy concerns, and intellectual property restrictions may prohibit full transparency. So we instead study the consequences of partial transparency—a common condition in real-world domains.

Bechavod et al. [2] study the effects of information discrepancy across different sub-populations of decision subjects on their ability to improve their observable features in strategic learning settings.

Like us, they do not assume the predictive model is fully known to the decision subjects. Instead, the authors model decision subjects as trying to infer the underlying predictive model by learning from their social circle of family and friends, which naturally causes different groups to form within the population. In contrast to this line of work, we study a setting in which the decision maker provides customized feedback to each decision subject individually. Additionally, while the models proposed by [17, 2] circumvent the assumption of full information about the deployed model, they restrict the decision subjects' knowledge to be obtained only through past data.

**Algorithmic recourse.** Our work is closely related to recent work on algorithmic recourse [28]. Algorithmic recourse is concerned with providing explanations and recommendations to individuals who have received unfavorable automated decisions. A line of algorithmic recourse methods including [43, 42, 25] focus on suggesting *actionable* or realistic changes to underlying qualifications to decision subjects interested in improving their decisions. Our action recommendations are "actionable" in the sense that they are interventions which promote long-term desirable behaviors while ensuring that the decision subject is not worse off in expectation.

**Transparency.** Recent legal and regulatory frameworks, such as the General Data Protection Regulation (GDPR) [10], motivate the development of forms of algorithmic transparency suitable for real-world deployment. While this work can be thought of as providing additional transparency into the decision-making process, it does not naturally fall into the existing organizations of explanation methods (e.g., as outlined in [7]), as our policy does not simply recommend actions based on the decision rule. Rather, our goal is to incentivize actionable interventions on the decision subjects' observable features which are desirable to the decision maker, and we leverage persuasion techniques to ensure compliance.

**Bayesian persuasion.** There has been growing interest in Bayesian persuasion [27] in the computer science and machine learning communities in recent years. Dughmi and Xu [12, 13] characterize the computational complexity of computing the optimal signaling policy for several popular models of persuasion. Castiglioni et al. [4] study the problem of learning the receiver's utilities through repeated interactions. Work in the multi-arm bandit literature [34, 33, 22, 6, 39] leverages techniques from Bayesian persuasion to incentivize agents to perform bandit exploration. Finally, linear programming-based approaches to Bayesian persuasion have been studied in the economics literature [30, 14], although the persuasion setting we study differs considerably.

**Other strategic learning settings.** The strategic learning literature [18, 17, 8, 32, 23, 2, 19, 20, 29, 16] broadly studies machine learning questions in the presence of strategic decision subjects. There has been a long line of work in strategic learning that focuses on how strategic decision subjects adapt their input to a machine learning algorithm in order to receive a more desirable prediction, although most prior work in this literature assumes that the underlying assessment rule is fully revealed to the decision subjects, which is typically not true in reality.

## 2 Setting and background

Consider a setting in which a decision maker assigns a predicted label $\hat{y} \in \{-1, +1\}$ (e.g., whether or not someone will repay a loan if granted one) to a decision subject with *initial* observable features $\mathbf{x}_0 = (x_{0,1}, \cdots, x_{0,d-1}, 1) \in \mathbb{R}^d$ (e.g., amount of current debt, bank account balance, etc.).[3] We assume the decision maker uses a fixed linear decision rule to make predictions, i.e., $\hat{y} = \text{sign}\{\mathbf{x}_0^\top \boldsymbol{\theta}\}$, where the assessment rule $\boldsymbol{\theta} \in \boldsymbol{\Theta} \subseteq \mathbb{R}^d$. The goal of the decision subject is to receive a positive classification (e.g., get approved for a loan). Given this goal, the decision subject may choose to take some *action* $a$ from some set of possible actions $\mathcal{A}$ to modify their observable features (for example, they may decide to pay off a certain amount of existing debt, or redistribute their debt to game the credit score). We assume that the decision subject has $m$ actions $\{a_1, a_2, \ldots a_m\} \in \mathcal{A}$ at their disposal in order to improve their outcomes. For convenience, we add $a_\emptyset$ to $\mathcal{A}$ to denote taking "no action". By taking action $a$, the decision subject incurs some *cost* $c(a) \in \mathbb{R}$. This could be an actual monetary cost, but it can also represent non-monetary notions of cost such as opportunity cost or the time/effort the decision subject may have to exert to take the action. We assume taking an action $a$ changes a decision subject's observable feature values from $\mathbf{x}_0$ to $\mathbf{x}_0 + \Delta \mathbf{x}(a)$, where $\Delta \mathbf{x}(a) \in \mathbb{R}^d$, and $\Delta \mathbf{x}_j(a)$ specifies the change in the $j$th observable feature as the result of taking

---

[3]We append a 1 to the decision subject's feature vector for notational convenience.

action $a$.[4] For the special case of $a_\emptyset$, we have $\Delta\mathbf{x}(a_\emptyset) = \mathbf{0}$, $c(a_\emptyset) = 0$. As a result of taking action $a$, a decision subject, *ds*, receives utility $u_{\text{ds}}(a, \boldsymbol{\theta}) = \text{sign}\{(\mathbf{x}_0 + \Delta\mathbf{x}(a))^\top \boldsymbol{\theta}\} - c(a)$. In other words, the decision subject receives some positive (negative) utility for a positive (negative) classification, subject to a *cost* for taking the action.

If the decision subject had exact knowledge of the assessment rule $\boldsymbol{\theta}$ used by the decision maker, they could solve an optimization problem to determine the best action to take in order to maximize their utility. However, in many settings it is not realistic for a decision subject to have perfect knowledge of $\boldsymbol{\theta}$. Instead, we model the decision subject's information through a *prior* $\Pi$ over $\boldsymbol{\theta}$, which can be thought of as "common knowledge" about the relative importance of various observable features in predicting the outcome of interest. For example, the decision subject may believe that prior payment history would likely be highly correlated with future default. We will use $\pi(\cdot)$ to denote the probability density function of $\Pi$ (so that $\pi(\boldsymbol{\theta})$ denotes the probability of the deployed assessment rule being $\boldsymbol{\theta}$). We assume the decision subject is rational and risk-neutral. So at any point during the interaction, if they hold a belief $\Pi'$ about the underlying assessment rule, they would pick an action $a^*$ that maximize their *expected* utility with respect to that belief. More precisely, they solve $a^* \in \arg\max_{a \in \mathcal{A}} \mathbb{E}_{\boldsymbol{\theta} \sim \Pi'}[u_{\text{ds}}(a, \boldsymbol{\theta})]$.

From the decision maker's perspective, some actions may be more desirable than others. For example, a bank may prefer that an applicant pay off more existing debt than less when applying for a loan. To formalize this notion of action preference, we say that the decision maker receives some utility $u_{dm}(a) \in \mathbb{R}$ when the decision subject takes action $a$. In the loan example, $u_{dm}(\text{pay off more debt}) > u_{dm}(\text{pay off less debt})$.

## 2.1 Bayesian persuasion in the algorithmic recourse setting

The decision maker has an *information advantage* over the decision subject, due to the fact that they know the true assessment rule $\boldsymbol{\theta}$, whereas the decision subject does not. The decision maker may be able to leverage this information advantage to incentivize the decision subject to take a more favorable action (compared to the one they would have taken according to their prior) by recommending an action to the decision subject according to a commonly known *signaling policy*.

**Definition 2.1** (Signaling policy). *A signaling policy $\mathcal{S} : \boldsymbol{\Theta} \to \mathcal{A}$ is a (possibly stochastic) mapping from assessment rules to actions.*[5]

We use $\sigma \sim \mathcal{S}(\boldsymbol{\theta})$ to denote the action recommendation sampled from signaling policy $\mathcal{S}$, where $\sigma \in \mathcal{A}$ is the realized recommended action.

The decision maker's signaling policy is assumed to be fixed and common knowledge. This is because in order for the decision subject to perform a Bayesian update based on the observed recommendation, they need to know the signaling policy. Additionally, the decision maker must have the *power of commitment*, i.e., the decision subject must believe that the decision maker will select actions according to their signaling policy. In our setting, this will be the case since the decision maker commits to their signaling policy before training the assessment rule. This can be seen as a form of transparency, as the decision maker is publicly announcing how they will use their assessment rule to provide action recommendations/recourse before they train the assessment rule. For simplicity, we assume that the decision maker shares the same prior beliefs $\Pi$ as the decision subject over the observable features before the model is trained. These assumptions are standard in the Bayesian persuasion literature (see, e.g., [27, 34, 33]).

In order for the decision subject to be incentivized to follow the actions recommended by the decision maker, the signaling policy $\mathcal{S}$ needs to be *Bayesian incentive-compatible*.

**Definition 2.2** (Bayesian incentive-compatibility). *Consider a decision subject* ds *with initial observable features $\mathbf{x}_0$ and prior $\Pi$. A signaling policy $\mathcal{S}$ is Bayesian incentive-compatible (BIC) for* ds *if $\mathbb{E}_{\boldsymbol{\theta} \sim \Pi}[u_{ds}(a, \boldsymbol{\theta})|\sigma = a] \geq \mathbb{E}_{\boldsymbol{\theta} \sim \Pi}[u_{ds}(a', \boldsymbol{\theta})|\sigma = a]$ for all actions $a, a' \in \mathcal{A}$ such that $\mathcal{S}(\boldsymbol{\theta})$ has positive support on $\sigma = a$.*

---

[4]Since we focus on a single decision subject, we hide the dependence of $\Delta\mathbf{x}$ on the initial feature value $\mathbf{x}_0$ to keep the notation simple.

[5]Note that since our model is focused on the decision maker's interactions with a single decision subject, we drop the dependence of $\sigma$ on the decision subject's characteristics.

---
**Example signaling policy** $\mathcal{S}(\theta)$

---

    **Case 1:** $\theta \in L$. Recommend action $a_1$ with probability $q$ and action $a_\emptyset$ with probability $1 - q$

    **Case 2:** $\theta \in M$. Recommend action $a_1$ with probability $1$

    **Case 3:** $\theta \in H$. Recommend action $a_1$ with probability $q$ and action $a_\emptyset$ with probability $1 - q$

---

Figure 2: Signaling policy for the example of Section 3.

In other words, a signaling policy $\mathcal{S}$ is BIC if, *given that the decision maker recommends action $a$*, the decision subject's expected utility is at least as high as the expected utility of taking any other action $a'$.

We remark that while for the ease of exposition our model focuses the interactions between the decision maker and a *single* decision subject, our results can be extended to a heterogeneous population of decision subjects—as long as we assume their interactions with the decision-maker are independent of one another (e.g., this assumption rules out one subject updating their belief based on the outcome of another subject's prior interaction with the decision-maker). Under such a setting, the decision maker would publicly commit to a method of computing the signaling policy, given a decision subject's initial observable features as input. Once a decision subject arrives, their feature values are observed and the signaling policy is computed.

## 3   Characterizing the utility gains of persuasion

As is generally the case in the persuasion literature [27, 26, 13], the decision maker can achieve higher expected utility with an optimized signaling policy compared to if they provided no recommendation or fully disclosed the model. To characterize *how much* leveraging the decision maker's information advantage may improve their expected utility under our setting, we study the following example.

Consider a simple setting under which a single decision subject has one observable feature $x_0$ (e.g., credit score) and two possible actions: $a_\emptyset =$ "do nothing" (i.e., $\Delta x(a_\emptyset) = 0$, $c(a_\emptyset) = 0$, $u_{dm}(a_\emptyset) = 0$) and $a_1 =$ "pay off existing debt" (i.e., $\Delta x(a_1) > 0$, $c(a_1) > 0$, $u_{dm}(a_1) = 1$), which in turn raises their credit score. For the sake of our illustration, we assume credit-worthiness to be a mutually desirable trait, and credit scores to be a good measure of credit-worthiness. We assume the decision maker would like to design a signaling policy to maximize the chance of the decision subject taking action $a_1$, regardless of whether or not the applicant will receive the loan. In this simple setting, the decision maker's decision rule can be characterized by a single threshold parameter $\theta$, i.e., the decision subject receives a positive classification if $x + \theta \geq 0$ and a negative classification otherwise. Note that while the decision subject does not know the exact value of $\theta$, they instead have some prior over it, denoted by $\Pi$.

Given the true value of $\theta$, the decision maker recommends an action $\sigma \in \{a_\emptyset, a_1\}$ for the decision subject to take. The decision subject then takes a possibly different action $a \in \{a_\emptyset, a_1\}$, which changes their observable feature from $x_0$ to $x = x_0 + \Delta x(a)$. Recall that the decision subject's utility takes the form $u_{ds}(a, \theta) = \mathrm{sign}\{((x_0 + \Delta x(a)) + \theta\} - c(a)$. Note that if $c(a_1) > 2$, then $u_{ds}(a_\emptyset, \theta) > u_{ds}(a_1, \theta)$ holds for any value of $\theta$, meaning that it is impossible to incentivize any rational decision subject to play action $a_1$. Therefore, in order to enable the decision maker to incentivize action $a_1$, we assume $c(a_1) < 2$.

We observe that in this simple setting, we can bin values of $\theta$ into three different "regions", based on the outcome the decision subject would receive if $\theta$ were actually in that region. First, if $x_0 + \Delta x(a_1) + \theta < 0$, the decision subject will not receive a positive classification, even if they take action $a_1$. In this region, the decision subject's initial feature value $x_0$ is "too low" for taking the desired action to make a difference in their classification. We refer to this region as $L$. Second, if $x_0 + \theta \geq 0$, the decision subject will receive a positive classification *no matter what* action they take. In this region, $x_0$ is "too high" for the action they take to make any difference on their classification. We refer to this region as $H$. Third, if $x_0 + \theta < 0$ and $x_0 + \Delta x(a_1) + \theta \geq 0$, the decision subject will receive a positive classification if they take action $a_1$ and a negative classification if they take action $a_\emptyset$. We refer to this region as $M$. Consider the signaling policy in Figure 2.

In Case 2, $\mathcal{S}$ recommends the action ($a_1$) that the decision subject would have taken had they known the true $\theta$, with probability 1. However, in Case 1 and Case 3, the decision maker recommends, with probability $q$, an action ($a_1$) that the decision subject would not have taken knowing $\theta$, leveraging the fact that the decision subject does not know exactly which case they are currently in. If the decision subject follows the decision maker's recommendation from $\mathcal{S}$, then the decision maker expected utility will increase from 0 to $q$ if the realized $\theta \in L$ or $\theta \in H$, and will remain the same otherwise. Intuitively, if $q$ is "small enough" (where the precise definition of "small" depends on the prior over $\theta$ and the cost of taking action $a_1$), then it will be in the decision subject's best interest to follow the decision maker's recommendation, *even though they know that the decision maker may sometimes recommend taking action $a_1$ when it is not in their best interest to take that action*. That is, the decision maker may recommend that a decision subject pay off existing debt with probability $q$ when it is unnecessary for them to do so in order to secure a loan. We now give a criteria on $q$ which ensures the signaling policy $\mathcal{S}$ is BIC.

**Proposition 3.1.** *Signaling policy $\mathcal{S}$ is Bayesian incentive-compatible if* $q = \min\{\frac{\pi(M)(2-c(a_1))}{c(a_1)(1-\pi(M))}, 1\}$, *where* $\pi(M) := \mathbb{P}_{\theta \sim \Pi}(x_0 + \theta < 0 \text{ and } x_0 + \Delta x(a_1) + \theta \geq 0)$.

*Proof Sketch.* We show that $\mathbb{E}_{\theta \sim \Pi}[u_{ds}(a_\emptyset, \theta) | \sigma = a_\emptyset] \geq \mathbb{E}_{\theta \sim \Pi}[u_{ds}(a_1, \theta) | \sigma = a_\emptyset]$ and $\mathbb{E}_{\theta \sim \Pi}[u_{ds}(a_1, \theta) | \sigma = a_1] \geq \mathbb{E}_{\theta \sim \Pi}[u_{ds}(a_\emptyset, \theta) | \sigma = a_1]$. Since these conditions are satisfied, $\mathcal{S}$ is BIC. The full proof may be found in Appendix C.

Next, we show that the decision maker's expected utility when recommending actions according to the optimal signaling policy can be *arbitrarily higher* than their expected utility from revealing full information or no information. We prove the following result in Appendix D.

**Proposition 3.2.** *For any $\epsilon > 0$, there exists a problem instance such that the expected decision maker utility from recommending actions according to the optimal signaling policy is $1 - \epsilon$ and the expected decision maker utility for revealing full information or revealing no information is at most $\epsilon$.*

## 4 Computing the optimal signaling policy

In Section 3, we show a one-dimensional example, where a signaling policy can obtain arbitrarily better utilities compared to revealing full information and revealing no information. We now derive the decision maker's optimal signaling policy for the general setting with arbitrary numbers of observable features and actions described in Section 2. Under the general setting, the decision maker's optimal signaling policy can be described by the following optimization:

$$
\max_{p(\sigma = a | \boldsymbol{\theta}), \forall a \in \mathcal{A}} \quad \mathbb{E}_{\sigma \sim \mathcal{S}(\boldsymbol{\theta}), \boldsymbol{\theta} \sim \Pi}[u_{dm}(\sigma)] \tag{1}
$$
$$
\text{s.t.} \quad \mathbb{E}_{\boldsymbol{\theta} \sim \Pi}[u_{ds}(a, \boldsymbol{\theta}) - u_{ds}(a', \boldsymbol{\theta}) | \sigma = a] \geq 0, \ \forall a, a' \in \mathcal{A},
$$

where we omit the valid probability constraints over $p(\sigma = a | \boldsymbol{\theta}), a \in \mathcal{A}$ for brevity. In words, the decision maker wants to design a signaling policy $\mathcal{S}$ in order to maximize their expected utility, subject to the constraint that the signaling policy is BIC. At first glance, the optimization may initially seem hopeless as there are infinitely many values of $p(\sigma = a | \boldsymbol{\theta})$ (one for every possible $\boldsymbol{\theta} \in \boldsymbol{\Theta}$) that the decision maker's optimal policy must optimize over. However, we will show that the decision maker's optimal policy can actually be recovered by optimizing over finitely many variables.

By rewriting the BIC constraints as integrals over $\boldsymbol{\Theta}$ and applying Bayes' rule, our optimization over $p(\sigma = a | \boldsymbol{\theta}), a \in \mathcal{A}$ takes the following form

$$
\max_{p(\sigma = a | \boldsymbol{\theta}), \forall a \in \mathcal{A}} \quad \mathbb{E}_{\sigma \sim \mathcal{S}(\boldsymbol{\theta}), \boldsymbol{\theta} \sim \Pi}[u_{dm}(\sigma)]
$$
$$
\text{s.t.} \quad \int_{\boldsymbol{\Theta}} p(\sigma = a | \boldsymbol{\theta}) \pi(\boldsymbol{\theta}) (u_{ds}(a, \boldsymbol{\theta}) - u_{ds}(a', \boldsymbol{\theta})) d\boldsymbol{\theta} \geq 0, \ \forall a, a' \in \mathcal{A}.
$$

Note that if $u_{ds}(a, \boldsymbol{\theta}) - u_{ds}(a', \boldsymbol{\theta})$ is the same for some "equivalence region" $R \subseteq \boldsymbol{\Theta}$ (which we formally define below), we can pull $u_{ds}(a, \boldsymbol{\theta}) - u_{ds}(a', \boldsymbol{\theta})$ out of the integral and instead sum over the different equivalence regions. Intuitively, an equivalence region can be thought of as the set of all $\boldsymbol{\theta} \in \boldsymbol{\Theta}$ pairs that are indistinguishable from a decision subject's perspective because they lead to the exact same utility for any possible action the decision subject could take. Based on this idea, we formally define a region of $\boldsymbol{\Theta}$ as follows.

**Definition 4.1** (Equivalence Region). *Two assessments $\boldsymbol{\theta}, \boldsymbol{\theta}'$ are equivalent (w.r.t. $u_{ds}$) if $u_{ds}(a, \boldsymbol{\theta}) - u_{ds}(a', \boldsymbol{\theta}) = u_{ds}(a, \boldsymbol{\theta}') - u_{ds}(a', \boldsymbol{\theta}')$, $\forall a, a' \in \mathcal{A}$. An equivalence region $R$ is a subset of $\boldsymbol{\Theta}$ such that for any $\theta \in R$, all $\theta'$ equivalent to $\theta$ are also in $R$. We denote the set of all equivalence regions by $\mathcal{R}$.*

For more intuition about the definition of an equivalence region, see Figure 5 in Appendix E. After pulling the decision subject utility function out of the integral, we can integrate $p(\sigma = a|\boldsymbol{\theta})\pi(\boldsymbol{\theta})$ over each equivalence region $R$. We denote $p(R)$ as the probability that the true $\boldsymbol{\theta} \in R$ according to the prior. Finally, since it is possible to write the constraints in terms of $p(\sigma = a|R)$, $\forall a \in \mathcal{A}, R \in \mathcal{R}$, it suffices to optimize directly over these quantities. For completeness, we include the constraints which make each $\{p(\sigma = a|R)\}_{a \in \mathcal{A}}, \forall R$ a valid probability distribution.

**Theorem 4.2** (Optimal signaling policy). *The decision maker's optimal signaling policy can be characterized by the following linear program OPT-LP:*

$$\max_{p(\sigma=a|R), \forall a \in \mathcal{A}, R \in \mathcal{R}} \quad \sum_{a \in \mathcal{A}} \sum_{R \in \mathcal{R}} p(R)p(\sigma = a|R)u_{dm}(a)$$

$$s.t. \quad \sum_{R \in \mathcal{R}} p(\sigma = a|R)p(R)(u_{ds}(a, R) - u_{ds}(a', R)) \geq 0, \ \forall a, a' \in \mathcal{A} \qquad \text{(OPT-LP)}$$

$$\sum_{a \in \mathcal{A}} p(\sigma = a|R) = 1, \ \forall R, \quad p(\sigma = a|R) \geq 0, \ \forall R \in \mathcal{R}, a \in \mathcal{A},$$

where $p(\sigma = a|R)$ denotes the probability of sending recommendation $\sigma = a$ if $\boldsymbol{\theta} \in R$. Note that the linear program OPT-LP is always feasible, as the decision maker can always recommend the action the decision subject would play according to the prior, which is BIC. Similarly, always recommending the action the decision subject would take had they known the assessment rule is also feasible.

While the problem of determining the decision maker's optimal signaling policy can be transformed from an optimization over infinitely many variables into an optimization over the set of finitely many equivalence regions $\mathcal{R}$, $|\mathcal{R}|$ may be exponential in the number of observable features $d$ (see Appendix F for more details). This is perhaps unsurprising as without any assumptions on $\boldsymbol{\Theta}$, the representation of the prior $\Pi$ can scale exponentially with $d$. In this case, we expect the running time of an algorithm which takes the entire prior as input to be exponential in the number of features as well. This motivates the need for a computationally efficient algorithm to approximate (OPT-LP), which does not require the full prior $\Pi$ as input.

We adapt the sampling-based approximation algorithm of Dughmi and Xu [13] to our setting in order to compute an $\epsilon$-optimal and $\epsilon$-approximate signaling policy in polynomial time, as shown in Algorithm 1 in Appendix G. At a high level, Algorithm 1 samples polynomially-many times from the prior distribution over the space of assessment rules, and solves an empirical analogue of (OPT-LP). In Appendix G, we show that the resulting signaling policy is $\epsilon$-BIC, and is $\epsilon$-optimal with high probability, for any $\epsilon > 0$. Formally, we prove the following statement.

**Theorem 4.3.** *Algorithm 1 runs in poly$(m, \frac{1}{\epsilon})$ time (where $m = |\mathcal{A}|$), and implements an $\epsilon$-BIC signaling policy that is $\epsilon$-optimal with probability at least $1 - \delta$.*

We leave open the question of whether there are classes of succinctly represented priors that permit efficient algorithms for computing the exact optimal policy in time polynomial in $d$ and $m$. It is also plausible to design efficient algorithms that only require some form of query access to the prior distribution. However, information-theoretic lower bounds of [13] rule out query access through sampling.

## 5   Experiments

In this section, we provide experimental results using a semi-synthetic setting where decision subjects are based on individuals in the Home Equity Line of Credit (HELOC) dataset [15]. The HELOC dataset contains information about 9,282 customers who received a Home Equity Line of Credit. Each individual in the dataset has 23 observable features related to an applicant's financial history (e.g., percentage of previous payments that were delinquent) and a label which characterizes their loan repayment status (repaid/defaulted). We compare the decision maker utility for different models of information revelation: our optimal signaling policy, revealing full information about the model, revealing no information about the model. As our theory predicts, the expected decision maker utility when recommending actions according to the optimal signaling policy either matches or exceeds the

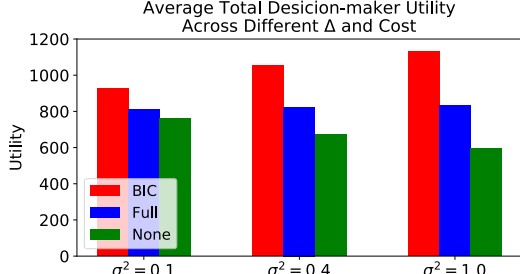

Average Total Desicion-maker Utility Across Different Δ and Cost

Figure 3: Total decision maker utility averaged across all cost and $\Delta \mathbf{x}(a)$ configurations for three different prior variances. The optimal signaling policy (red) consistently yields higher utility compared to the two baselines: revealing full information (blue) and no information (green). This gap increases when the decision subject is less certain about the model being used (higher $\sigma^2$).

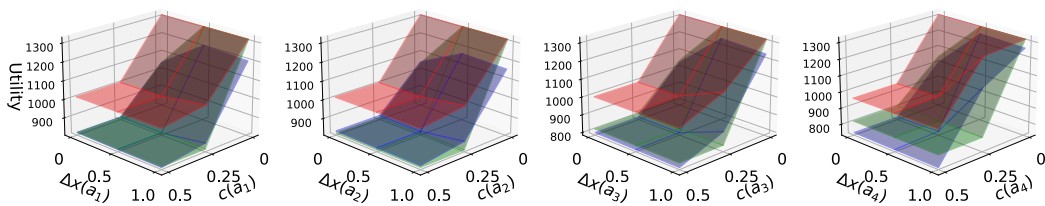

Figure 4: Expected utility across different $c(a)$ and $\Delta \mathbf{x}(a)$ configurations for $\sigma^2 = 0.4$. Optimal signaling policy (red) effectively upper-bounds the two baselines, revealing everything (blue) and revealing nothing (green) in all settings.

expected utility from revealing full information or no information about the assessment rule across all problem instances. Moreover, the expected decision maker utility from signaling is *significantly* higher on average. Next, we explore how the decision maker's expected utility changes when action costs and changes in observable features are varied jointly. Our results are summarized in Figures 3 and 4.

In order to adapt the HELOC dataset to our strategic setting, we select four features and define five hypothetical actions $\mathcal{A} = \{a_\emptyset, a_1, a_2, a_3, a_4\}$ that decision subjects may take in order to improve their observable features. Actions $\{a_1, a_2, a_3, a_4\}$ result in changes to each of the decision subject's four observable features, whereas action $a_\emptyset$ does not. For simplicity, we view actions $\{a_1, a_2, a_3, a_4\}$ as equally desirable to the decision maker, and assume they are all more desirable than $a_\emptyset$. Using these four features, we train a logistic regression model that predicts whether an individual is likely to pay back a loan if given one, which will serve as the decision maker's realized assessment rule. For more information on how we constructed our experiments, see Appendix I.

Given a $\{(c(a_i), \Delta \mathbf{x}(a_i))\}_{i=1}^4$ instance and information revelation scheme, we calculate the decision maker's total expected utility by summing their expected utility for each applicant. Figure 3 shows the average total expected decision maker utility across different $\Delta \mathbf{x}(a)$ and cost configurations for priors with varying amounts of uncertainty. See Figure 9 in Appendix I.3 for plots of all instances which were used to generate Figure 3. Across all instances, the optimal signaling policy (red) achieves higher average total utility compared to the other information revelation schemes (blue and green). The difference is further amplified whenever the decision subjects are less certain about the true assessment rule (i.e., when $\sigma$ is large). Intuitively, this is because the decision maker leverages the decision subjects' uncertainty about the true assessment rule in order to incentivize them to take desirable actions, and as the uncertainty increases, so does their ability of persuasion.

To better understand how the decision maker's expected utility changes as a function of $c(a)$ and $\Delta \mathbf{x}(a)$, we sweep through multiple $\{(c(a_i), \Delta \mathbf{x}(a_i))\}_{i=1}^4$ tuples on a grid of $(c(a_i), \Delta \mathbf{x}(a_i)) \in \{0, 0.25, 0.5\} \times \{0, 0.5, 1.0\}$ for $i \in \{1, 2, 3, 4\}$ and measure the effectiveness of the three information revelation schemes. Figure 4 shows the surface of the decision maker utility as a function of $(c(a_i), \Delta \mathbf{x}(a_i))$ for the optimal signaling policy (red), revealing full information (blue), and revealing no information (green). When $c(a_i)$ is high and $\Delta \mathbf{x}(a_i)$ is low, the total expected decision maker utility is low as there is less incentive for the decision subject to take actions (although even under this setting, the optimal signaling policy outperforms the other two baselines). As $c(a_i)$ decreases and $\Delta \mathbf{x}(a_i)$ increases, the total expected decision maker utility increases.

# 6 Conclusion

We investigate the problem of offering algorithmic recourse without requiring full transparency (i.e., revealing the assessment rule). We cast this problem as a game of Bayesian persuasion, and offer several new insights regarding how a decision maker can leverage their information advantage over decision subjects to incentivize mutually beneficial actions. Our stylized model relies on several simplifying assumptions, which suggest important directions for future work:

**Public persuasion.** We assume that the recommendations received by each decision subject are *private*. However, if a decision subject is given access to recommendations for multiple individuals, it may be possible for them to reconstruct the underlying model. While out of the scope of this work, it would be interesting to study models of *public* persuasion in the algorithmic recourse setting.

**Beyond linear decision rules.** We focus on settings with *linear* decision rules and assume all decision subject parameters (e.g., cost function, initial observable features, etc.) are known to the decision maker. We leave it for future work to extend our findings to non-linear decision rules, or settings in which some of the decision subjects' parameters are unknown to the decision maker.

# 7 Acknowledgements

KH is supported by a NDSEG Fellowship. ZSW and KH were supported in part by the NSF FAI Award #1939606, a Google Faculty Research Award, a J.P. Morgan Faculty Award, a Facebook Research Award, and a Mozilla Research Grant. AT was supported in part by the National Science Foundation grants IIS1705121, IIS1838017, IIS2046613, IIS2112471, a funding from Meta, Morgan Stanley and Amazon. HH acknowledges support from NSF IIS2040929, a CyLab 2021 grant, and a Meta (Facebook) research award. Any opinions, findings and conclusions or recommendations expressed in this material are those of the author(s) and do not necessarily reflect the views of any of these funding agencies. KH would like to thank Haifeng Xu for insightful conversations about Dughmi and Xu [13]. KH would also like to thank James Best, Yatong Chen, Jeremy Cohen, Daniel Ngo, Chara Podimata, and Logan Stapleton for helpful suggestions and conversations.

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
