# OpenReview forum: "Bayesian Persuasion for Algorithmic Recourse"
_NeurIPS.cc/2022/Conference — NeurIPS 2022 Accept_

### Official Review · Reviewer_WZjo · 2022-07-06

**Rating:** 6
**Confidence:** 4
**Soundness:** 3 good
**Presentation:** 3 good
**Contribution:** 3 good

**Summary:**

The paper studies a setting of algorithmic recourse in the classification setting, where the decision subjects present strategic behavior. The authors propose a model, using the framework of Bayesian persuasion, where a decision maker and a decision subject have prior beliefs about the parameters of a predictive model. The decision maker first commits to a form of transparency (signaling policy), trains the predictive model and communicates to a decision subject which action to take in order to change their features in a way that the model gives them a positive prediction. The decision subject, based on the recommendation, adapts their posterior belief about the model parameters and takes an action that maximizes their expected utility. The goal of the paper is to compute a signaling policy that maximizes the decision maker's expected utility under the constraint of "Bayesian incentive compatibility", i.e., that it is in the best interest of the decision subject to take the action recommended to them by the decision maker. The authors propose an approximation algorithm for the problem, based on linear programming, and evaluate their methodology using semi-synthetic data.

**Questions:**

My 2 main concerns/questions regarding the assumptions of the proposed model that I would like the authors to discuss are the following:
1. What is the real need to assume that the decision maker has a prior over the model parameters $\theta$? They have the data, they train the model. Where is the uncertainty about $\theta$ coming from if the training process is completely under their control? More technically, in the problem definition of Section 4, why does the decision maker sample $\theta$ from $\Pi$? Related to that, why would the decision maker commit to a signaling policy before training? These look to me like assumptions needed to make the whole thing fit to the Bayesian persuasion setting but they slightly disregard how algorithmic recourse would work in real-life applications. I think these assumptions need to be better motivated in the text.
2. Why is the decision maker's utility assumed to be a function of the action $a$ and not a function of $x_0 +\Delta x(a)$? If it is independent of $x_0$, then the decision maker gets the same utility from two decision subjects taking the same action but with one of them crossing the threshold and the other one not crossing it. This sounds a bit weird to me. Is a completely untrustworthy loan applicant who pays some of their existing debt the same as a borderline candidate who repays some of their existing debt? How is it natural that the final output of the classifier doesn't contribute to the decision maker's utility?

**Limitations:**

I think the authors sufficiently discuss the limitations of their methodology, specifically that it focuses solely on linear models and it doesn't consider shared information between different decision subjects. The societal implications section is also ok.

**Strengths And Weaknesses:**

Originality: To the best of my knowledge, this is the first paper framing algorithmic recourse as an instance of Bayesian persuasion. This connection is a novel and conceptually interesting contribution. The paper contains a quite comprehensive related work section discussing work on strategic machine learning, algorithmic recourse and bayesian persuasion. Although the technical setup is different, there is some relevant work (see [1]) connecting counterfactual explanations (concept almost equivalent to algorithmic recourse) with strategic machine learning and also tries to compute personalized recommendations that maximize the decision maker's utility. I believe it should be cited and discussed in the "strategic responses to unknown predictive models" subsection.

Quality: I found the paper technically sound. I read part of the proofs in the Appendix and they appear to be correct. The framing of  algorithmic recourse as an instance of Bayesian persuasion and the overall formulation is mostly reasonable. However, there are certain assumptions of the model that I am not sure to what extent they reflect the real-life problem that the paper is studying. I elaborate more on these assumptions in the "Questions" section. Finally, the experimental evaluation is satisfying but I think that it could have been a bit more extensive. My main concern related to the experiments is that the authors assume all the available actions give equal utility to the decision maker and they are all more desirable than the null action that doesn't change the decision subject's features. However, a large part of the strategic machine learning literature is studying malicious adaptation which is performed to "trick" the decision maker and therefore, in the current setup, would lead to negative utility. As far as I understand, the authors' framework could capture this by considering a set of actions with a mixture of negative and positive utilities. I believe that, in that case, the results of Figure 1 might have been different. For example, I think that "Full information" could lead to lower utility than "No information" and it would be interesting to see how the method proposed by the authors would perform in comparison.

Clarity: The paper is nicely written and easy to follow. I have a few minor suggestions for improvement which I list below.
1. Starting in line 202, the authors discuss the assumption that the decision maker publishes and commits to a signaling policy. However, the arguments presented are trying to fit algorithmic recourse to the Bayesian persuasion setting rather than explaining how the interaction described in the table of page 2 would work in practice. It is not clear to me what it means for a decision maker to publish a signaling policy in lending, college admissions, hiring e.t.c. I think the authors could improve this paragraph by adding some real-life examples.
2. There is a full stop punctuation mark missing in line 206.
3. In lines 210-211 the authors talk about "prior beliefs $\Pi$ over the observable features" which kind of implies that $\Pi$ is a belief about the distribution of features $x$. However, as far as I understand, $\Pi$ is a belief about the model parameters $\theta$. I think the aforementioned phrase is causing confusion.
4. There is a citation number missing in line 331.
5. I would encourage the authors to bring Algorithm 1 to the main body of the paper. The authors provide a high-level description of it but, since the algorithm itself is not super complicated and it is their main technical tool towards solving their problem, it shouldn't be left for the appendix.
6. In lines 335 & 337, the authors mention that the algorithm leads to an $\epsilon$-BIC signaling policy, however, only the definition of a BIC policy has been given in the paper so far and $\epsilon$-BIC is undefined.
7. In the experimental section, it would be useful if the authors presented in detail their baselines "full information" and "no information". To my understanding, these two correspond to the decision subject best-responding based on a posterior $\Pi'$ which is (i) a point mass on $\theta$ and (ii) equal to the prior $\Pi$. If that is correct, I think the authors should write it explicitly when describing the experimental setup.
8. There is a missing reference in line 652 (Appendix F).

Significance: I think the paper carries interesting ideas regarding the general problem of algorithmic recourse, namely that the decision subjects have prior beliefs about the decision rule and that the decision maker can provide recommendations to incentivize beneficial long-term outcomes (e.g., repaying existing debts instead of manipulating financial records in the credit scoring example). Therefore, the conceptual contributions are important and the accompanying methodology is sound.

[1] Tsirtsis, Stratis, and Manuel Gomez Rodriguez. "Decisions, counterfactual explanations and strategic behavior." Advances in Neural Information Processing Systems 33 (2020): 16749-16760.

POST REBUTTAL
-------------------------------------
I read the authors response and the authors addressed most of my concerns about the proposed model. I keep my initial score, in favor of accepting the paper.

---

> ### Author Response · Authors · 2022-08-01
> **Reply to Reviewer WZjo (1/2)**
>
> Thanks for your thorough review and helpful comments. Please find our responses below.
>
> >Although the technical setup is different, there is some relevant work (see [1]) connecting counterfactual explanations (concept almost equivalent to algorithmic recourse) with strategic machine learning and also tries to compute personalized recommendations that maximize the decision maker's utility. I believe it should be cited and discussed in the "strategic responses to unknown predictive models" subsection.
>
> Thanks for pointing out this omission. We will include a comparison with this work in the revision.
>
> >Finally, the experimental evaluation is satisfying but I think that it could have been a bit more extensive. My main concern related to the experiments is that the authors assume all the available actions give equal utility to the decision maker and they are all more desirable than the null action that doesn't change the decision subject's features.
>
> Our assumption that all available actions give equal utility to the decision subject was made for convenience. We reran the experiments with different utility values for each action and our overall findings were the same: our optimal signaling policy achieves higher average total utility compared to baselines. See section I.4 in the appendix for these additional experimental results.
>
> >a large part of the strategic machine learning literature is studying malicious adaptation which is performed to "trick" the decision maker and therefore, in the current setup, would lead to negative utility. As far as I understand, the authors' framework could capture this by considering a set of actions with a mixture of negative and positive utilities. I believe that, in that case, the results of Figure 1 might have been different. For example, I think that "Full information" could lead to lower utility than "No information" and it would be interesting to see how the method proposed by the authors would perform in comparison.
>
> We focus on the setting of strategic (not adversarial) decision subjects (e.g., someone who is applying for a loan cares about receiving the loan, not adversarially harming the performance of the deployed model). However, our model would be able to handle this case by suitably modifying the decision subject's utility function (although this may result in a different definition of "equivalence region" than the one we currently use). Since the underlying assumptions about decision subject behavior are different, we agree that our empirical findings in such a setting could be different.
>
> Reply to Question 1:
>
> >What is the real need to assume that the decision maker has a prior over the model parameters θ? They have the data, they train the model. Where is the uncertainty about θ coming from if the training process is completely under their control?
>
> The decision maker has uncertainty about the model because they commit to their signaling policy before the model is trained (and its true value is revealed). While we view the concurrent design of the decision rule and signaling policy as an interesting direction for future work, we would like to point out that this is often not possible under many settings for either practical or institutional reasons. In the case of our running example on lending, a credit scoring agency may be in charge of determining the assessment rule, not the bank offering the loan.
>
> >More technically, in the problem definition of Section 4, why does the decision maker sample θ from Π?
>
> The process of training the model itself can be viewed as sampling the model from a distribution. For example, the process of training the model on i.i.d. sampled data will result in a distribution over models.
>
> >Related to that, why would the decision maker commit to a signaling policy before training?
>
> The decision maker may commit to a signaling policy before training due to various transparency concerns; for example, either due to regulation, or the desire to build trust with their decision subjects, even when the model itself cannot be revealed.
>
> >These look to me like assumptions needed to make the whole thing fit to the Bayesian persuasion setting but they slightly disregard how algorithmic recourse would work in real-life applications. I think these assumptions need to be better motivated in the text.
>
> While we disagree that these assumptions somewhat disregard how algorithmic recourse would work in real-life applications (and we hope our answers to the previous part of this question have convinced you otherwise), we agree that these assumptions could be better motivated and we plan on expanding upon them in the revision.

---

> > ### Author Response · Authors · 2022-08-01
> > **Reply to Reviewer WZjo (2/2)**
> >
> > Reply to Question 2:
> >
> > >Why is the decision maker's utility assumed to be a function of the action a and not a function of x0+Δx(a)?
> >
> > While out of the scope of this work, we believe that allowing for more general models of decision maker utility is an interesting direction for future research. In the context of our running example, the reviewer is correct that a bank may care whether or not someone who receives a loan repays it. However, paying off some existing debt would be desirable to the bank regardless of whether or not the applicant receives a loan, if the applicant's debt is to the bank they are applying to. More generally, we believe it is natural for the decision maker to have some preference over the actions taken by the decision subject, and we chose the decision maker utility to reflect this.

---

> > > ### Comment · Reviewer_WZjo · 2022-08-05
> > > **Thanks for your clarifications**
> > >
> > > Thanks for taking the time to address the negative points in the review. One last question and one suggestion:
> > > 1. Perhaps I wasn't clear but when I mentioned "malicious adaptation which is performed to "trick" the decision maker" I wasn't referring to adversarial agents who would try to harm the classifier and, therefore, would have a different utility function. In the context of the utility function presented in the paper, I was wondering how the model captures actions taken by the agents that try to "game" the classifier, e.g., loan applicants who manipulate their records in order to receive a loan. In the model, would that correspond to a set of actions with mixed positive (repaying debt) and negative (manipulating records) utilities? Or is it equivalent to the experimental setup in Appendix I.4 where all actions have positive (but different) utilities?
> > > 2. The explanations about the model assumptions and how it fits to the setting of algorithmic recourse were helpful. I think that the paper would benefit significantly by a real-world running example (e.g., on lending or hiring) explaining all the steps in the table of page 2, e.g, where does the uncertainty about $\theta$ come from, what is the signaling policy and why the decision maker commits to it before training, e.t.c.

---

> > > > ### Author Response · Authors · 2022-08-05
> > > > **Reply to Reviewer WZjo's response**
> > > >
> > > > Thanks for your suggestion. We agree that a real-world running example would be helpful to motivate our model, and we will include such a running example in our revision.
> > > >
> > > > Regarding your first point, our model can indeed capture so-called “gaming” actions (e.g., manipulating records) by assigning non-positive decision maker utilities to them as you suggest.

---

### Official Review · Reviewer_x6ZH · 2022-07-10

**Rating:** 7
**Confidence:** 3
**Soundness:** 3 good
**Presentation:** 3 good
**Contribution:** 2 fair

**Summary:**

This work explores the Algorithmic Recourse problem through the lens of Bayesian Persuasion. The recourse problem is cast as a persuasion problem, where the Decision Maker/Principal is the operator of machine learning system (represented as a classifier), and the Decision Subject/Agent is the consumer of prediction. Motivating example is online banking, where the decision subject applies for a loan, and the classifier decides if they are eligible.

Classifiers are assumed to be linear  $y=\mathrm{sign}(\theta^T x)$, where $x$ is the feature vector, and the weights $\theta$ are assumed to be distributed according to a common prior $\theta \sim \Pi$. For the interaction protocol, the decision maker first commits to a signaling scheme which is the agent action to recommend as a function of model weights $\theta$. After the signaling scheme is set, an instance $\theta \sim \Pi$ is picked at random and revealed to the decision maker, the signaling scheme is applied, and the decision subject takes the rationally-optimal action according to their posterior belief. In this persuasion setting, actions $a\in\mathcal{A}$ are modeled as changes to the feature vector, such that $x’=x+\Delta(a)$ when action $a$ is taken. Each action is also assumed to entail a cost $c(a)$.

Authors present three sets of results. The first result attempts to illustrate the importance of taking persuasion considerations into account by presenting an instance of the problem in which the utility discrepancy between naive and strategic decision maker behavior can be arbitrarily close to the maximal discrepancy (Proposition 3.2). The second set of results discusses the computational complexity of the problem, and presents an efficient sampling-based algorithm which calculates an approximately optimal policy in $poly(m,\frac{1}{\epsilon})$ time (Theorem 4.3). The algorithm requires a polynomial number of samples of $\theta \sim \Pi$. The third set of results is an empirical evaluation using the HELOC lending dataset.

**Questions:**

* Realistic model assumptions - One possible way to model the randomness in model parameters $\theta$ is assuming that $\theta$ is the result of an Empirical Risk Minimization process on a dataset of feature-label pairs $(x_1,y_1),\dots,(x_n,y_n)$ sampled iid from a distribution $\mathcal{D}$. If we assume this, does a common prior assumption on $\theta$ entail equivalent assumption in terms of $\mathcal{D}$? In other words - If we assume that $\theta$ is obtained using ERM, is the common prior assumption on $\theta$ equivalent to assuming that both Principal and Agent have the ability to train the prediction model themselves?
* What happens if the same value of $\theta$ is reused across many instances of the persuasion?
* What are the main insights from the set of empirical evaluations? Can we design an experiment which tests the method under more realistic assumptions? I guess that both positive or negative results will be interesting in this context.
* Is it possible to add components to the mechanism, or describe a setting in which the decision maker is likely to adhere to their commitment, and don't examine any data privately before committing to the signaling scheme?
* Algorithm 1 assumes that $\theta$ can be sampled polynomially-many times. Assuming that sampling from $\Pi$ can be very costly, is it possible to trade off running time for lower sample complexity?


**Limitations:**

Main limitation of this work from my perspective is the modeling assumptions, which may not hold in many realistic use cases. I think these should be outlined as a basis for further discussion. See details above.


**Strengths And Weaknesses:**

Strengths
* Problem is well-motivated. From the theoretical perspective, it is interesting to find new domains where the Bayesian Persuasion perspective can be useful. From the practical perspective, strategic response and recourse are becoming increasingly prevalent in real-world systems.
* Writing is clear and easy to follow, mathematical limitations of results are clearly illustrated.
* Work brings up interesting directions for discussion and future work.

Weaknesses
* Some of the core assumptions made by the model are not realistic. In particular, assuming that a common prior exists for $\theta$ seems highly non-trivial - For example, decision makers usually have a resource advantage, and for example are likely to conduct a market survey revealing more details about $\Pi$ before committing to a signaling scheme - Breaking the common prior assumption. Moreover, model training in the real world is often very costly, so it is more realistic to assume that model parameters $\theta$ will be reused many times, and obtaining fresh samples $\theta \sim \Pi$ (e.g as assumed by the presented algorithm) will be very costly. In addition, in many realistic settings users have the ability to share data and possibly collude.
* Not sure whether this model naturally extends beyond linear classification and one-shot settings. As an extreme example - Will it be realistic to assume that both parties have a common prior over the parameters of a modern, large-scale neural network?
* It seems to me that the empirical evaluation section mainly provides a numerical validation for the theoretical results, and does not explore much beyond them. As the theoretical model relies on non trivial assumptions, the experiments section can be an opportunity to explore limitations and robustness.

---

> ### Author Response · Authors · 2022-08-01
> **Reply to Reviewer x6ZH (1/2)**
>
> Thanks for your detailed review. Please find our responses below.
>
> Weaknesses:
>
> >Some of the core assumptions made by the model are not realistic. In particular, assuming that a common prior exists for θ seems highly non-trivial - For example, decision makers usually have a resource advantage, and for example are likely to conduct a market survey revealing more details about Π before committing to a signaling scheme - Breaking the common prior assumption.
>
> While the common prior assumption is standard within the Bayesian persuasion literature, the reviewer is correct that many real-world settings exist in which this assumption may be unrealistic. We view this as an interesting and important direction for future work, but we would like to point out that if the decision maker has privileged information such that their prior is better informed than the decision subject’s, it is possible for them to use this to induce a common prior (although doing so may or may not be in their best interest). In the context of our running example on lending, if the decision maker/bank has a resource advantage, they could reveal information from market surveys, etc. on their website such that both the decision maker and decision subjects have the same information (thus inducing the same prior). We will further clarify this point in the next revision of our submission.
>
> >Moreover, model training in the real world is often very costly, so it is more realistic to assume that model parameters θ will be reused many times, and obtaining fresh samples θ∼Π (e.g as assumed by the presented algorithm) will be very costly. In addition, in many realistic settings users have the ability to share data and possibly collude.
>
> The reviewer is correct that in many settings users have the ability to share data and collude. As we mention in the conclusion, we view this as an important direction for future research.
>
> >Not sure whether this model naturally extends beyond linear classification and one-shot settings. As an extreme example - Will it be realistic to assume that both parties have a common prior over the parameters of a modern, large-scale neural network?
>
> While a common prior assumption over neural network parameters is indeed unrealistic, a more pressing issue in this example is the lack of interpretability of complex models like neural networks. Appropriately explaining these models in an actionable manner is a separate but active area of research. Because of their inherent lack of interpretability, decision subjects may have difficulty reasoning about different outcomes if decisions are made using deep neural networks, therefore limiting the effectiveness of persuasion (even if a common prior assumption were to hold). However, extending our results to other interpretable ML models (e.g. decision trees) seems possible and would be interesting.
>
> >It seems to me that the empirical evaluation section mainly provides a numerical validation for the theoretical results, and does not explore much beyond them. As the theoretical model relies on non trivial assumptions, the experiments section can be an opportunity to explore limitations and robustness.
>
> We would like to point out that we do vary several parameters of the model in our experiments (e.g., cost of actions, how much actions affect changes in observable features - see Figure 2), and show that persuasion has an effect across a wide variety of settings. However, we agree it would be interesting to further relax these assumptions.
>
> Questions:
>
> >Realistic model assumptions - One possible way to model the randomness in model parameters θ is assuming that θ is the result of an Empirical Risk Minimization process on a dataset of feature-label pairs (x1,y1),…,(xn,yn) sampled iid from a distribution D. If we assume this, does a common prior assumption on θ entail equivalent assumption in terms of D? In other words - If we assume that θ is obtained using ERM, is the common prior assumption on θ equivalent to assuming that both Principal and Agent have the ability to train the prediction model themselves?
>
> A common prior assumption on the decision rule could indeed be obtained by a common prior assumption on the distribution of data used to train the model, as long as both the principal and agent agree on the specifics of how the model is trained (e.g., number of training samples, model hyperparameters, etc.)
>
> >What happens if the same value of θ is reused across many instances of the persuasion?
>
> Since our results are in terms of expected utility (where the expectation is with respect to the prior over the decision rule), they hold for both the setting in which the decision rule is trained once and reused multiple times, and the setting in which the model is repeatedly retrained using different data.

---

> > ### Author Response · Authors · 2022-08-01
> > **Reply to Reviewer x6ZH (2/2)**
> >
> > >What are the main insights from the set of empirical evaluations? Can we design an experiment which tests the method under more realistic assumptions? I guess that both positive or negative results will be interesting in this context.
> >
> > The main insight from our empirical evaluations is that persuasion is beneficial across a wide range of strategic decision making settings (i.e. our results are not dependent on specific x(a) and c(a) values). We agree that conducting experiments under more realistic settings (e.g., running mechanical turk experiments) would be interesting.
> >
> > >Is it possible to add components to the mechanism, or describe a setting in which the decision maker is likely to adhere to their commitment, and don't examine any data privately before committing to the signaling scheme?
> >
> > Our current model of interaction assumes that the decision maker has the power to commit to a signaling scheme, and adhere to their commitment. In practice, this may be enforceable via laws or regulations. However, it would be interesting to relax this assumption in future work.
> >
> > >Algorithm 1 assumes that θ can be sampled polynomially-many times. Assuming that sampling from Π can be very costly, is it possible to trade off running time for lower sample complexity?
> >
> > There is a trade off between accuracy and number of samples, as the number of samples required for an epsilon approximation grows as 1/epsilon^2. Therefore if the accuracy threshold is decreased, the required number of samples from the distribution (and therefore the runtime of the algorithm) will decrease.

---

> > > ### Comment · Reviewer_x6ZH · 2022-08-09
> > > **Response**
> > >
> > > Thank you for your response, and I appreciate your feedback. Following a review of the authors' responses, my rating remains unchanged. Despite making very strong assumptions that may not be realistic, this model utilizes Bayesian Persuasion in a novel manner to address a well-motivated problem. Thus, I consider it a step in the right direction, and I believe it has the potential to influence future research in this area.

---

### Official Review · Reviewer_qh5D · 2022-07-10

**Rating:** 5
**Confidence:** 4
**Soundness:** 3 good
**Presentation:** 3 good
**Contribution:** 2 fair

**Summary:**

The paper studies an application of Bayesian persuasion to a linear classification model. In this model, a decision maker uses a linear decision rule to make a decision on a subject, which is represented by a feature vector. The subject is uncertain about the specific rule the decision maker uses to reach the decision but only has a prior belief over it. The decision maker can then strategically reveal this information to persuade the subject to take certain actions. The goal of the study is to design an algorithm to compute the optimal persuasion strategy, so that the decision maker's utility is maximized.

**Questions:**

As described above, it would be great if the authors could explain why the decision maker does not directly optimize the distribution of $\theta$ but choose to use signaling.

**Limitations:**

The authors did not seem to discuss any potential nagative social impact. I think there is the possibility that banks or financial institutions use their information advantage to manipulate customers to act in favor of them but not in favor of the social welfare.

**Strengths And Weaknesses:**

The model is original. The idea of applying Bayesian persuasion to the linear classification model is interesting. The paper is well-writen and clear overall. The thecnical results look somewhat standard and unsurprising, following mostly the standard approach to solving the private signaling problem.

My major concern is about the setup of the model. In the model, the decision maker selects both the state (i.e. decision rule) and the signaling strategy. This is different from typical Bayesian persuasion models where the decision maker has no control over the state. Crucially, this raises the question whether it is indeed useful to use signaling in such a model. In other words, why doesn't the decision maker just optimize their strategy about how to choose the decision rule (i.e. optimize the distribution of $\theta$) - I think they can well just do this to acheive the same (or even a better) outcome without using signaling. (Analogously, in Stackelberg games, the leader does not benifit from signaling when they already have the power to make a commitment.)

The authors didn't seem to have justified this setup adequately in the paper. I tried to come up with a justification but found it hard to justify the model if the decision maker has control over $\theta$ but chooses to _not_ optimize the distribution of it. So it seems that this setup is justifiable only in the case where the decision maker cannot choose the decision rule. Nevertheless, this seems to deviate from the motivating examples (eg. bank and a customer applying for a loan): one particular question is if the decision maker does not choose the rule, then who does that in these examples? Notice that this is different than saying that the decision maker cannot control the prior belief of the subject - even when the decision maker cannot control the subject's prior belief, they can still optimize their actual selection of $\theta$.

---

> ### Author Response · Authors · 2022-08-01
> **Reply to Reviewer qh5D**
>
> Thanks for your review. Please find our replies to your concerns below.
>
> >My major concern is about the setup of the model. In the model, the decision maker selects both the state (i.e. decision rule) and the signaling strategy. This is different from typical Bayesian persuasion models where the decision maker has no control over the state. Crucially, this raises the question whether it is indeed useful to use signaling in such a model. In other words, why doesn't the decision maker just optimize their strategy about how to choose the decision rule (i.e. optimize the distribution of θ) - I think they can well just do this to acheive the same (or even a better) outcome without using signaling.
>
> We emphasize that the set up we consider is solely focused on designing the signaling policy when the decision rule is exogenously determined. The reviewer raises the concern that this setup is not adequately justified or well-motivated in the context of algorithmic decision making. While we view the joint optimization of the decision rule and signaling policy as an interesting technical direction for future work, we would like to point out that this is often not possible in many real-world decision making settings, for either practical or institutional reasons. In the context of our running example on lending, a credit scoring agency, not the bank offering the loan, may be in charge of determining the assessment rule used to evaluate decision subjects. Another interaction commonly discussed when making decisions in the presence of strategic agents is one in which a teacher (principal) interacts with a student (strategic agent). See, e.g. [1,2] for papers which consider this setting. Under such a setting, the teacher may not be in charge of designing the evaluation of the student, but may still have knowledge of the exam. For example, the evaluation may be designed by some government agency, as is the case with many standardized tests in the United States, and the teacher may be given access to the exam in advance, or have knowledge of it from previous years of teaching. Under such a setting, the teacher may still wish to offer the student some way of improving their chances of success. While the teacher cannot directly reveal the evaluation to the student, they can recommend actions (e.g., topics to study) in order to give the student a chance to improve themselves. This setting is also captured by our model. We will make this motivation more clear in the introduction.
>
> >The authors did not seem to discuss any potential nagative social impact. I think there is the possibility that banks or financial institutions use their information advantage to manipulate customers to act in favor of them but not in favor of the social welfare.
>
> We would like to point out that we do discuss potential negative social impacts in Appendix A (this is mentioned in the checklist). However, we agree with the reviewer that a discussion of potential negative social impacts is important for this type of work. We will move this discussion to the main body of the paper.
>
> [1]: Jon Kleinberg and Manish Raghavan. How do classifiers induce agents to invest effort strategically? ACM Transactions of Economics and Computation (EC), 2019.
>
> [2]: Keegan Harris, Hoda Heidari, and Zhiwei Steven Wu. Stateful Strategic Regression. Neural Information Processing Systems (NeurIPS), 2021.

---

> > ### Comment · Reviewer_qh5D · 2022-08-08
> > **Thanks for your reply**
> >
> > Thank you for your reply. I think in your paper (and in the revision) it says that the assessment rule $\boldsymbol\theta$ is chosen by the decision maker (around line 160). The critical issue here is that if the signal sender has control over $\boldsymbol\theta$, then they can well just optimize $\boldsymbol\theta$ to induce a desirable action of the receiver, and there is no need or benefit to further use a signaling strategy on top of that, or to optimize it jointly with $\boldsymbol\theta$. So indeed, optimizing $\boldsymbol\theta$ is the more important thing for a decision-maker to do in such circumstances. Optimizing both of them is unnecessary and somewhat meaningless. And optimizing only the signaling strategy while treating $\boldsymbol\theta$ as exogenously determined is in some sense a misleading solution as it may be suboptimal compared with optimizing $\boldsymbol\theta$. I think this is the case for all the examples mentioned in the paper, in particular the ones in footnote 1, where it is obviously completely up to the decision-maker to choose the assessment rule. So I still don't see a convincing motivation or any benefit to use information about the decision rule to persuade the subject in these examples. Even in the first example in your reply, I think it is still the bank, instead of the credit scoring agency, who selects the decision rule to decide whether to give a loan to the applicant.

---

> > > ### Author Response · Authors · 2022-08-08
> > > **Clarifying the motivation for our setting**
> > >
> > > >I think in your paper (and in the revision) it says that the assessment rule θ is chosen by the decision maker (around line 160).
> > >
> > > Thanks for pointing this out. We realize that this choice of wording may be confusing to the reader, and we will update this passage in the revision. The setting we study is one in which joint optimization of the decision rule and signaling policy is not possible. If such joint optimization were allowed, we agree that the use of persuasion would be less meaningful.
> > >
> > >
> > > However, we disagree that it is “obviously completely up to” the person/entity offering recourse to choose the assessment rule in the examples mentioned in our submission (hiring, college admissions, lending). Oftentimes, the decision maker (e.g., a bank) is not a single person or entity. In reality, different entities within the decision making institution may be responsible for different aspects of the decision making process. In hiring, a recruiter for a company may have knowledge of the factors the company uses to make hiring decisions. While the recruiter may not be allowed to reveal this information (e.g., for fear of lawsuits, see https://smallbusiness.chron.com/companies-give-reasons-didnt-hire-20141.html), they may still wish to offer the candidates they recruit some way of increasing their chances of being hired. In lending, one department of the bank may be in charge of determining the threshold on the credit assessment, while someone else may be in charge of offering recourse. Similar logic applies to the college admissions example, in which someone associated with the university may have the ability to offer advice to applicants, but does not have the ability to unilaterally change the underlying assessment rule.
> > >
> > >
> > > In the revision, we will clarify how these examples fit within our setting. Additionally, we will include the student/teacher example from our previous reply, in which it may be more immediately apparent how and why the assessment rule is exogenously determined.

---

> > > > ### Comment · Reviewer_qh5D · 2022-08-09
> > > > **more comments**
> > > >
> > > > I think there is a distinction. In your model, the sender chooses to not disclose full information not because they are not allowed to but because they are better off not doing that. This are no restrictions on how much information the decision-maker can disclose in your model, and the case with such restrictions would require a different model, so I think hiring or college admission may not be good examples. The student/teacher example seems a better fit. But it's better to set up the teacher's goal as somewhat different from the student's - otherwise, why doesn't the teacher just reveal full information unless there are restrictions on how much they can reveal (but again with restrictions you may need to a different model). I'm upgrading my score given this example but strongly encourage the authors to clarify the problem setting and motivations in the revision (preferably, provide better examples).

---

> ### Author Response · Authors · 2022-08-05
> **Follow-up Message to Reviewer qh5D**
>
> Please let us know if our response has sufficiently addressed your concerns regarding the setup of our model and potential negative social impacts. We would be happy to answer any other questions you may have.

---

### Official Review · Reviewer_iKBd · 2022-07-11

**Rating:** 5
**Confidence:** 3
**Soundness:** 3 good
**Presentation:** 3 good
**Contribution:** 3 good

**Summary:**

This paper models a strategic interaction between a decision maker and a decision subject in an incomplete information game using Bayesian persuasion, which can be applied to credit scoring.  Specifically, the utilities of both the decision maker and decision subject depend on the action of each other, while their true decision rules are not public information. Moreover, there is an action recommendation system designed by the decision maker to incentivize the decision subject to modify their actions. In their model, the authors show that the decision maker can design an incentive-compatible recommendation system such that the modified action of the decision subject will benefit both participants. Furthermore, they develop an algorithm to approximately build this system within polynomial time. Finally, they also use numerical experiments to illustrate the benefits of their approach.

**Questions:**

1. I have a question about the extension to multiple decision subjects. When there is a heterogeneous group of decision subjects, will the decision maker assign the same threshold parameter $\theta$ for all participants or assign different parameters from the same distribution?

2. Another question is about manipulations, which is mentioned in the introduction as follows ''The question we are interested in answering in this work is: how can the decision maker incentivize decision subjects to take such beneficial actions while discouraging manipulations?". I wonder whether the utility function, actions, and related rewards and consumptions of the decision subject are public information available to the decision maker. If it is true, the decision subject cannot make profits based on private knowledge. Thus, this model might not be able to reflect manipulations by the decision subject.


3. For the numerical experiments part, what is the difference between the obtained utility of the decision subject in your model and that in the full information model?

**Limitations:**

I think there is no potential negative social impact of the work.

**Strengths And Weaknesses:**

Strengths:

1. This is the first work to use Bayesian persuasion to model the interaction described in the summary part based on the previous literature.

2. In their model, the authors show the existence of an incentive-compatible action recommendation system.

3. The authors develop an approach to construct this system and another time-efficient algorithm to approximately design the desired system in polynomial time.

Weaknesses:

1. The model only contains one decision maker and one single decision subject. Although the authors state that this model can be extended to a heterogeneous case with heterogeneous decision subjects, I still have one question listed in the question part below.

2. The authors may have implicitly assumed that the utility function, actions, and corresponding consequences of the decision subject are known to the decision maker when designing the algorithm. Please see bullet 2 in the question part for more detail.

---

> ### Author Response · Authors · 2022-08-01
> **Response to Reviewer iKBd**
>
> Thanks for your review. Since your mentioned weaknesses of our work are all listed as questions, we only address the questions.
>
> >When there is a heterogeneous group of decision subjects, will the decision maker assign the same threshold parameter θ for all participants or assign different parameters from the same distribution?
>
> Our results hold in expectation over the distribution on θ. Thus, our results are applicable to both the setting in which the decision maker uses the same decision rule for each decision subject and the setting in which the decision maker uses different decision rules drawn from the same distribution. We note, however, that in some domains, the decision-maker may be bound to apply the same rule to all subjects.
>
> >I wonder whether the utility function, actions, and related rewards and consumptions of the decision subject are public information available to the decision maker. If it is true, the decision subject cannot make profits based on private knowledge. Thus, this model might not be able to reflect manipulations by the decision subject.
>
> The reviewer is correct that, in our model, the decision maker knows the decision subject’s set of actions and utility function, while the decision subject does not have any private knowledge. We view addressing the setting where the decision subject has private knowledge as an interesting extension for future work. However, we would like to emphasize that the decision subject having private information is not a prerequisite for having the ability to pick an action which maximizes their utility in expectation.
> Furthermore, we would like to emphasize that the main focus of our work is to study settings in which the decision rule is unknown to the decision subjects, as full information about the decision rule being used is an unrealistic assumption made by many works studying high-stakes decision making (see, e.g., [18,19,20,23,29] our submission).
>
> >For the numerical experiments part, what is the difference between the obtained utility of the decision subject in your model and that in the full information model?
>
> The “full information” results refer to a particular policy instantiation in which the decision maker completely reveals the assessment rule to the decision subject. We include this policy as a baseline because most standard “strategic learning” work (e.g., [18,19,20,23,29]) assumes the decision subject has complete knowledge of the assessment rule.

---

> > ### Comment · Reviewer_iKBd · 2022-08-07
> > **Reply to the Rebuttal**
> >
> > Thank you for the response. My questions are almost addressed. One follow-up question is as follows.
> >
> > In the first part, let me rephrase my question. The optimal policy is a set of conditional probabilities which depends on the parameter $\theta$. It means that the decision must depend on the parameter. If there is a heterogeneous group of decision subjects, my question is whether their parameters are revealed to the decision maker. If so, I think this assumption is too strong to apply the result.

---

> > > ### Author Response · Authors · 2022-08-08
> > > **Clarifying the multiple decision subject setting**
> > >
> > > Thanks for your reply. The set up we consider is focused on designing the optimal signaling policy when the decision rule (θ) is exogenously determined. Therefore, the optimal policy is a set of conditional probabilities for any realization of θ, which is always revealed to the decision maker but may or may not differ between decision subjects.  As we mention in our previous reply,
> > >
> > > >Our results hold in expectation over the distribution on theta. Thus, our results are applicable to both the setting in which the decision maker uses the same decision rule for each decision subject and the setting in which the decision maker uses different decision rules drawn from the same distribution. We note, however, that in some domains, the decision-maker may be bound to apply the same rule to all subjects.
> > >
> > > If this has not sufficiently addressed your question, we would be happy to elaborate further.

---

### Meta-Review · Area_Chair_hvZW · 2022-08-24

**Recommendation:** Accept
**Confidence:** Certain

**Metareview:**

The paper formulates the problem of algorithmic recourse under partial transparency as a Bayesian persuasion game. It is shown that the decision-maker can design an incentive-compatible action signaling strategy with guarantees that both the decision-maker and decision-subjects are not worse off in terms of expected utility. The results provide several insights into the complexity of computing an optimal signaling strategy;  moreover, a polynomial-time approximation algorithm is provided to compute a near-optimal signaling strategy. The reviewers acknowledged that the paper considers an important problem setting and provides new technical insights into algorithmic recourse using the framework of Bayesian persuasion. However, the reviewers also raised several concerns and questions in their initial reviews. We want to thank the authors for their detailed responses and for actively engaging with the reviewers during the discussion phase. The reviewers appreciated the responses, which helped in answering their key questions. The reviewers have an overall positive assessment of the paper, and there is a consensus for acceptance. The reviewers have provided detailed feedback in their reviews, and we strongly encourage the authors to incorporate this feedback when preparing the final version of the paper.

**Award:**

No

---

### Decision · Program_Chairs · 2022-09-14

Accept